# Translation, reliability, and validity of Amharic versions of the Pelvic Floor Distress Inventory (PFDI-20) and Pelvic Floor Impact Questionnaire (PFIQ-7)

**Tadesse Belayneh Melkie**[1]*, **Zelalem Mengistu Gashaw**[2], **Zelalem Ayichew Workineh**[2], **Tamiru Minwuye Andargie**[2], **Tibeb Zena Debele**[3], **Solomon Gedlu Nigatu**[4]

1 Department of Anesthesia, School of Medicine, College of Medicine and Health Sciences, University of Gondar, Gonda, Ethiopia, 2 Department of Obstetrics and Gynecology, School of Medicine, College of Medicine and Health Science, University of Gondar, Gonda, Ethiopia, 3 Department of Clinical Midwifery, School of Midwifery, College of Medicine and Health Sciences, University of Gondar, Gonda, Ethiopia, 4 Department of Epidemiology and Biostatistics, Institute of Public Health, College of Medicine and Health Sciences, University of Gondar, Gonda, Ethiopia

* tadbel20@gmail.com

**Data Availability Statement:** All relevant data are within the paper and its Supporting information files.

## Abstract

### Purpose

Pelvic Floor Disorders (PFDs) affects many women and have a significant impact on their quality of life. Pelvic Floor Impact Questionnaire (PFIQ-7) and Pelvic Floor Distress Inventory (PFDI-20) help to assess PFDs; however, both are not culturally translated into the Amharic-language. Hence, we aimed to translate the English versions of short forms of the PFDI-20 and PFIQ-7 into Amharic-language and evaluate their psychometric properties in Amharic-speaking Ethiopian women with symptomatic PFDs.

### Methods

The PFDI-20 and PFIQ-7 were translated into Amharic language using standard procedures. The Amharic versions were completed by 197 patients (response rate 92%) with PFDs from University of Gondar specialized and comprehensive Hospital. Internal consistency and test-retest reliability were examined through Cronbach's alpha and Intraclass correlation coefficients (ICC). A relative criterion standard, POP-SS-7 score, was correlated with total PFDI-20 and subscale POPDI-6 scores using spearman's rank order correlation (SCC). Construct validity was evaluated by known group validity using the Mann–Whitney U test.

### Results

Both instruments were successfully translated and adapted with an excellent content validity (> 0.90). The Amharic versions of the PFDI-20 and PFIQ-7 showed excellent internal consistency and test-retest reliability in both summary and subscales (Cronbach's alpha: 0.92 for PFDI-20 and 0.91 for PFIQ-7; and ICC: 0.97 for PFDI-20 and 0.86 for PFIQ-7). Criterion

**Funding:** This work was supported by a grant, R. No: V/P/RCS/05/540/2020, from the University of Gondar. The University was not involved in the design, conduct, analysis or interpretation of the study, or the review or approval of the manuscript.

**Competing interests:** The authors have declared that no competing interests exist.

validity was good for POPDI-6 (SCC = 0.71; p < 0.001). Moreover, construct validity was acceptable, showing significant differences among groups of PFDs in the PFDI-20 and PFIQ-7 scores (Mann–Whitney U Test; p < 0.001).

## Conclusions

The Amharic versions of the PFDI-20 and PFIQ-7 are comprehensible, reliable, valid, and feasible in Ethiopian Amharic-speaking women with PFDs to evaluate symptoms and its impact during research and clinical practice. However, further studies are needed to evaluate the responsiveness.

## Introduction

Pelvic floor disorders (PFDs), one of the chronic women's health problems, is a condition that comprises pelvic organ prolapse (POP), urinary incontinence (UI), and fecal incontinence (FI) [1]. Globally PFDs are common debilitating health problem that affects several women [2]. In low and middle-income countries (LMICs), the prevalence of one or more PFDs has been reported to be 25% [3]. In Ethiopia, although many women with PFDs do not disclose their health problem [4], one-fifth of women suffers from at least one type of PFDs [5].

Although mortality is rare, PFDs negatively impact women's health-related quality of life (HRQoL) with physical, psychological, sexual, and social implications [2, 3, 6, 7]. The severity of symptoms and their impact on QoL are important parameters in the management and follow-up of women with PFDs [8, 9]. Hence, in urogynecology, the use of patient-reported outcomes (PRO) for outcome measures is recommended [10]. PROs are measures of health reported directly by patients without amendment or interpretation by clinicians or anyone else [11].

In women with PFDs, several PRO measures have been developed for measuring symptoms burden, functional status and QoL [12]. The Pelvic Floor Distress Inventory (PFDI-20) and the Pelvic Floor Impact Questionnaire (PFIQ-7) are the two most frequent, generic, psychometrically valid, and reliable PROs for measuring the extent to which PFDs affect HRQoL [13, 14]. PFDI investigates the range of PFD symptoms and the inconvenience they cause, while PFIQ covers the impact on daily life. Both instruments are widely used and have been translated into several different countries and languages [15–18]. In Ethiopia, PFDI-20 and PFIQ-7 have also been validated in Tigrigna [19]. However, they are not adapted and validated for Amharic-speaking Ethiopian women. In contrast, there are POP-specific Amharic versions (the Prolapse Symptom Score (POP-SS) and Prolapse Quality of Life (P-QoL-20)) that measure POP symptoms, their severity, and their impact on HRQoL [20, 21]. Amharic is the working language of federal government in Ethiopia with millions of native and second-language speakers. It is spoken as the first language in the region where the study was conducted [22, 23]. It is clear that before the use of any PROs in a different language, they should always be adapted and their psychometric characteristics tested in the new language [24]. This helps to reduce flawed research conclusions and allows comparison of international studies [25]. Therefore, we aimed to translate the PFDI-20 and PFIQ-7 into the Amharic language and evaluate their measurement properties (reliability and validity).

## Materials and methods

The study was conducted in two phases: phase 1 included the translation and adaptation of the instruments into the Amharic language and phase 2 comprised the evaluation of their

psychometric properties. The University of Gondar Institutional Ethics Review Committee approved the study protocol (Registration number: V/P/RCS/05/540/2020; December 15, 2020). Also, translation permission was obtained from the original instrument developers. All participants were informed and gave their written consent before participating in the study.

## Measurement instruments

The PFDI consists of 20 questions and contains three subscales: Pelvic Organ Prolapse Distress Inventory (POPDI-6) with six questions about the inconvenience of the prolapse, Colorectal-Anal Distress Inventory (CRADI-8) with eight questions concerning difficulties of defecation, and the Urinary Distress Inventory (UDI-6) with six questions about difficulties in urination. Response options for rating distress associated with each symptom ranged from 0 ("no" as in no symptoms) to 1 ("not at all" as in symptoms are present but not bothered at all) to 4 ("quite a bit" as in symptoms are present and quite a bit bothered). The mean score of answered items is multiplied by 25 to obtain the scale score (range 0–100) on each scale. Summary scores are calculated by adding up the scale scores (range 0–300). Higher scores indicate more symptom distress [13, 14].

The PFIQ consists of 21 items that measures the impact of bladder, bowel, and vaginal symptoms on daily activity, social/relationships, and emotional health. It has three scales: the Pelvic Organ Prolapse Impact Questionnaire (POPIQ-7), the Colorectal-Anal Impact Questionnaire (CRAIQ-7), and the Urinary Impact Questionnaire (UIQ-7). Response options range from 0 ("not at all") to 3 ("quite a bit"). In each scale, the mean score of answered items is multiplied by 33.3 to obtain the scale score (range 0–100). Summary scores are calculated by adding up the scale scores (range 0–300). Higher scores indicate more impact on daily activity [13, 14].

The POP-SS has seven items that focus on symptoms caused or aggravated by prolapse [26]. Each question requires participants to rate the frequency of symptom experienced in the four weeks prior to evaluation. Symptom responses were rated on a 5-point Likert scale (0 = never, 1 = somewhat, 2 = sometimes, 3 = most of the time, and 4 = always). The total score ranged from 0 to 28. The total score was calculated by summating scores for individual symptom responses [26]. In this study the Amharic version was used and the detail was found elsewhere [20].

## Linguistic translation and adaptation to Amharic language

We followed multistep translation method [27] including forward translation, discussion with translators, back-ward translation, expert discussion, and cognitive test on purposively selected patients. First, to ensure semantic equivalence, the English versions were independently translated into Amharic by two native experts who are fluent in the English-language, one of them was familiar with the study subject. These two translated versions were reviewed by the members of the researchers to create an initial Amharic versions. Next, the initial Amharic versions were then back-translated into English by two independent native English speakers (they are from the University of Gondar English language department) fluent in Amharic who were blinded and naïve to the English versions. Thirdly, a one-day session was held to examine the forward-and-back-translated versions with the original versions by urogynecologists, urologist, a colorectal surgeon, and a psychometrician. Experts evaluate whether instruments were equivalent in grammar, meaning, concept, and quality of the overall translation. They examine all of the proceeding steps and selected the most appropriate translation for each item or provided alternate translations. The overall discussion and meeting with experts were led by the primary investigator (TB). At this stage, minor inconsistencies between the original and the

translated versions were resolved. Finally, to determine the readability of the language used, cultural context, and to ask for their feedback on the comprehensiveness of items cognitive debriefing was conducted with a convenience sample of ten native Amharic-speaking women with and/or without PFDs visiting the gynecology clinic of the University of Gondar Comprehensive and Specialized Hospital (UoGCSH). During cognitive debriefing, women were asked to say something on the overall comprehensiveness of the tool, and some modifications in some words were done. Minor problems were identified without the need to adapt the content of the questionnaires. After the amendment of minor discrepancies, the Amharic versions were then finalized for the evaluation of the psychometric properties of the scales.

## Psychometric evaluation

A cross-sectional study was conducted to examine the psychometric properties of the Amharic versions of the PFDI-20 and PFIQ-7 between November 2020 and August 2021. Women with and without POP were recruited from the gynecology outpatient clinics of the UoGCSH. Women who had a psychiatric problem/cognitive impairment, could not speak or understand Amharic, had undergone previous pelvic and/or anti-incontinence surgery within 6 months preceding the study, had a known or suspected current pregnancy, were postpartum (the first 6 weeks following childbirth) or <18 years of age were excluded.

The sample size was determined based on the recommendations of at least 5 subjects per item by the Consensus-based Standards for the Selection of Health Measurement Instruments (COSMIN) [28]. To this end, the estimated sample was 205. But we included 10 participants to protect against dropout and missing responses. Then the final participants we approached were 215.

The participants (n = 215) completed the paper form of the Amharic versions of PFDI-20 and PFIQ-7 at baseline. Fifty-seven patients were randomly selected for test-retest analysis to complete the same questionnaires again two weeks later. Patients were selected if there were no symptomatic changes and no interventions were taken. The interval is considered short enough to avoid changes in presenting symptoms and long enough for patients to forget their previous responses). This follow-up assessment was performed with face-to-face interviews by the same data collectors who collected the baseline.

After completing the instruments at baseline, all participants underwent a gynecological examination in the dorsolithotomy position and the prolapse stage was classified using the Pelvic Organ Prolapse Quantification System (POP-Q) [29] by the research team members who were blinded to the questionnaires score of the particular patient. Accordingly, women were assigned to stage 0 with no prolapse. Stage I, leading point of the wall of the vagina or cervix remains at least 1 cm above the hymenal ring. Stage II, leading point descends to the introitus, defined as an area extending from 1 cm above to 1 cm below the hymenal ring. The leading point descends >1 cm outside the hymenal ring in stage III. However, it does not form a complete vaginal vault eversion or procidentia uteri. Stage IV, complete vaginal vault eversion or procidentia uteri [29]. Moreover, the definitions of the International Continence Society were used to describe the clinical symptoms associated with PFDs [30, 31]. Thus, a woman was considered to have UI if the following conditions were confirmed. Stress urinary incontinence (SUI): when there was involuntary leakage of urine on activities like effort or exertion, or sneezing or coughing; urge incontinence (UUI): when there was involuntary leakage that goes along with or immediately preceded by urgency; and mixed urinary incontinence (MUI): when there was involuntary leakage of urine associated with both urgency and stress UI. AI was defined by self-report of gas, stool, both gas and stool. FI was defined as at least one symptom of incontinence of flatus or stool (only flatus, only loose stool, only normal stool, or the combination of flatus, loose, and normal stool) [30, 31].

## Statistical analysis

All statistical analyses were performed using SPSS for Windows 20.0 (SPSS Inc., Chicago, IL, USA). A P-value of <0.05 was considered statistically significant. Participants' demographic and clinical characteristics were summarized using descriptive statistics (percentages, frequencies, means, and standard deviations). Normality was assessed for the outcome variables using the Shapiro-Wilk test. The COSMIN recommendations were used for evaluating the psychometric properties. Methodological testing, including reliability and validity was assessed [28].

Reliability was evaluated using internal consistency and test-retest reliability. Cronbach's alpha was calculated to determine the internal consistency of subscale (correlation between the items within a subscale) and summary (overall correlation between the items within a scale) scores in the PFDI-20 and PFIQ-7 questionnaires, and values above 0.7 were considered adequate [28]. Test-retest reliability, the degree to which a measurement is free of error, was computed using the interclass correlation coefficients (ICC) over a two-week interval [32]. Women were asked if their condition had changed during the interim period to evaluate their stability using the question: 'Compared to the first time you completed the questionnaires, has your PFD condition changed?' (If 'Yes', women were excluded from the retest). We assumed there would be no real change in a women's level of function within two-week intervals. Good values of test-retest reliability were considered greater than 0.70 [28].

Validity was evaluated using content, criterion and construct validity. Content validity (the degree to which the content of a measure is an adequate reflection of the target construct), was assessed by an expert panel and cognitive debriefing interviews of women during the abovementioned Amharic translation process. Experts were asked to evaluate each item for content validity on a four-point Likert scale (1 = not relevant, 2 = somewhat relevant, 3 = quite relevant, 4 = highly relevant). Thereafter, their agreement was calculated using Content Validity Index (CVI). A CVI of ≥0.80 was considered acceptable [33]. The expert panel consisted of two urogynecologists, and one urologist, colorectal surgeon, physiotherapist, and psychometrician. Moreover, we calculated the floor and ceiling effects as a component of content validity. We considered problematic if more than 15% of participants achieved the highest or lowest possible score [28].

Criterion validity, which describes how well the questionnaire correlates with an existing gold standard, was evaluated by comparing a relative criterion standard, the POP-SS-7 score, with total PFDI-20 and subscale POPDI-6 scores using Spearman's correlation coefficient (SCC). The SCC was defined as 0.8–1.0 excellent, 0.61–0.80 very good, 0.41–0.60 good, 0.21–0.40 sufficient, and 0.00–0.20 poor [34].

Construct validity was evaluated by known group validity and hypothesis testing. For known group validity, patients were categorized into two groups: With POP (POP-Q score > 2) and without POP, With UI and without UI, With AI and without AI. We hypothesized that PFDI-20 and PFIQ-7 scores are correlated with POP-Q scores, and patients with higher scores of POP-Q had bothersome symptoms (PFDI) and poor quality of life (PFIQ). Moreover, patients with POP would have higher POPDI-6 scores than those without these symptoms. Patients with UI had higher UDI-6 scores than those without those symptoms. Summary and subscale PFDI-20 and PFIQ-7 scores of these groups were tested using the Mann-Whitney U test since these scores did not follow a normal distribution. Construct validity was considered adequate when at least 75% of these hypotheses were confirmed [28].

To evaluate the PFDI-20 and PFIQ-7 Amharic version's feasibility, the percentage of unanswered individual items and the percentage of patients who did not answer any of the items were analyzed. Also, the average administration time was calculated.

## Results

### Characteristics of participants

Of 234 consecutive women, 215 (91.9%) were eligible for inclusion of which 197 (91.6%) consented to participate at baseline. Women's average age was 44 ± 14 years and the mean parity was 5.1 ± 3.3.

Among those 197 participants at baseline 117 (59.4%) were diagnosed with some form of POP (stage I-IV), of which 96 (48.8%) had POP-Q stage III and IV. Eighty had no symptoms of PFDs. Among women with POP, 63 had symptoms of UI and 36 had AI. Of the 63 women with UI, 36 had SUI, 23 had UUI, and the remaining two had mixed UI. Among the 36 women with AI, 21 had flatulence only, 12 had FI only, and the remaining three had both. The characteristics of the study participants are presented in Table 1.

At baseline, mean scores of the total PFDI-20 and PFIQ-7 were 116.4 ± 48.4 and 92.2 ± 34.2, respectively, in women with PFDs. Similarly, in women without PFDs, mean scores were 2.2 ± 3.6 and 1.6 ± 2.3 for the PFDI-20 and PFIQ-7, respectively (Table 2).

### Translation of the questionnaires

Both forward-and-backward translation were performed as planned and no major problem was encountered. The revisions made by the experts and women in the pilot study guaranteed the content validity of the Amharic versions of the PFDI-20 and PFIQ-7. In the pilot study, most of the items were well understood by the participating women and confirmed face validity. No words or items showed difficulty in their comprehensibility and did not need to be

**Table 1. Characteristics of study participants, University of Gondar, Ethiopia (n = 197).**

| Characteristic | Statistics |
|---|---|
| Age (years) Mean ± SD | 44.6 ± 13.6 |
| Parity Mean ± SD | 5.1 ± 3.3 |
| Mode of delivery, n (%) | |
| Instrument | 4 (2.2) |
| Vaginal | 145 (80.6) |
| CS | 15 (8.3) |
| Both vaginal and CS | 7 (3.9) |
| POP-Q findings, n (%) | |
| Stage 0 | 80 (40.6) |
| Stage I | 5 (2.5) |
| Stage II | 16 (8.1) |
| Stage III | 60 (30.5) |
| Stage IV | 36 (18.3) |
| Urinary incontinence, n (%) | |
| SUI | 38 (19.3) |
| UUI | 23 (11.7) |
| MUI | 2 (1.0) |
| Anal incontinence, n (%) | |
| Fecal | 12 (6.1) |
| Flatulence | 21 (10.7) |
| Fecal and flatulence | 3 (1.5) |

*SD* standard deviation, *CS* cesarean section *POP-Q* Pelvic Organ Prolapse Quantification, *SUI* stress urinary incontinence, *UUI* urge urinary incontinence, *MUI* mixed urinary incontinence

**Table 2. Pelvic Floor Distress Inventory–20 and Pelvic Floor Impact Questionnaire-7 scale and summary scores at baseline, University of Gondar, Ethiopia.** (n = 197).

| Characteristic | With PFD (n = 117) | Without PFD (n = 80) |
|---|---|---|
| PFDI-20 | 116.4 ± 48.4 | 2.2 ± 3.6 |
| POPDI-6 | 53.1 ± 22.2 | 2.9 ± 4.6 |
| CRADI-8 | 18.4 ± 12.8 | 1.1 ± 2.3 |
| UDI-6 | 40.2 ± 25.0 | 2.2 ± 4.5 |
| PFIQ-7 | 92.2 ± 34.2 | 1.6 ± 2.3 |
| UIQ-7 | 47.6 ± 23.4 | 1.1 ± 2.8 |
| CRAIQ-7 | 10.5 ± 7.2 | 1.2 ± 2.1 |
| POPIQ-7 | 60.6 ± 24.2 | 1.1 ± 1.9 |

*PFDI-20* Pelvic Floor Distress Inventory–Short Form 20, *POPDI* Pelvic Organ Prolapse Distress Inventory, *CRADI* Colorectal–Anal Distress Inventory, *UDI* Urinary Distress Inventory, *PFIQ-7* Pelvic Floor Impact Questionnaire-7, *UIQ* Urinary Impact Questionnaire, *CRAIQ* Colorectal–Anal Impact Questionnaire, *POPIQ* Pelvic Organ Prolapse Impact Questionnaire.

adapted. The expert panel found the questionnaires, questions and rating scale clinically reasonable and relevant for the setting. The final versions of the Amharic PFDI-20 and PFIQ-7 maintain the structure of the original versions. The Amharic and original English versions of PFDI-20 and PFIQ-7 are shown in (S1 and S2 Files).

## Evaluation of measurement properties

**Feasibility.** Concerning feasibility, all women responded to all items in the Amharic PFDI-20 and PFIQ-7 versions, and no missing items were found. Data collectors reported no difficulties in asking the items and no women reported having met problems in understanding the items. The average time for questionnaire administration was 9.4 min for PFDI-20 and 7.2 min for PFIQ-7.

**Reliability.** Internal consistency and test–retest reproducibility are shown in Table 3. Cronbach's alpha coefficient values obtained from the Amharic versions of the PFDI-20 and

**Table 3. Reliability of the Amharic versions of the PFDI-20 and PFIQ-7 questionnaires.**

| | Test–retest Reliability (n = 50) | | Internal consistency (n = 197) |
|---|---|---|---|
| | ICC | P value[a] | Cronbach's Alpha |
| PFDI-20 | 0.97 | <0.001 | 0.92 |
| POPDI-6 | 0.96 | <0.001 | 0.88 |
| CRADI-8 | 0.95 | <0.001 | 0.84 |
| UDI-6 | 0.96 | <0.001 | 0.91 |
| PFIQ-7 | 0.86 | <0.001 | 0.91 |
| UIQ-7 | 0.90 | <0.001 | 0.93 |
| CRAIQ-7 | 0.87 | <0.001 | 0.95 |
| POPIQ-7 | 0.94 | <0.001 | 0.93 |

*ICC* Intraclass Correlation. *PFDI-20* Pelvic Floor Distress Inventory–Short Form 20, *POPDI* Pelvic Organ Prolapse Distress Inventory, *CRADI* Colorectal–Anal Distress Inventory, *UDI* Urinary Distress Inventory, *PFIQ-7* Pelvic Floor Impact Questionnaire-7, *UIQ* Urinary Impact Questionnaire, *CRAIQ* Colorectal–Anal Impact Questionnaire, *POPIQ* Pelvic Organ Prolapse Impact Questionnaire

[a] Single rating, absolute agreement, and a two-way mixed-effects model.

PFIQ-7 total scores were 0.92 (95% CI: 0.91, 0.94) and 0.91 (95% CI: 0.88, 0.92), respectively, demonstrating excellent internal consistency. Similarly, their subscale coefficients also showed excellent internal consistency, with values varying from 0.84 to 0.95 (Table 3).

To assess test-retest reliability, 57 women were reinterviewed two-weeks later. Seven women reported a change in POP severity and were removed from the test-retest analysis, and 50 women completed this test-retest correctly. As shown in Table 3, the test-retest reliability was excellent between the paired scores of the total and subscale scores of PFDI-20 and PFIQ-7. The ICC was 0.97 in the PFDI-20 total score, and a range from 0.95 to 0.96 was found for its subscales. Moreover, the total PFIQ-7 score showed an ICC of 0.86, and its subscales varied from 0.87 to 0.94. In both cases, all the ICCs were statistically significant ($p < 0.001$) (Table 3).

## Content validity

Content validity was established through expert committee review during the process of adaptation and qualitative analysis of women's comments during a pilot study. The multidisciplinary expert panel agreed that the questionnaires included all the relevant items, and no questions were added to the original versions. The mean CVI result for the PFDI-20 was 0.90 and for the PFIQ-7 was 0.92, indicating acceptable content validity. The interviewed women indicated that all items were well understood, and that the questionnaires showed good readability and comprehensibility.

## Floor effect and ceiling effect

No ceiling effects were observed in either group on the PFDI-20 and PFIQ-7 summary and subscale scores. However, a floor effect was seen in both scales and subscales from both instruments, with minimum frequency varying from 29.4% to 39.1% on PFDI-20 and 26.9% to 40.6% on PFIQ-7. In women with PFD, a floor effect was seen on the PFDI-20 (29.9%) and PFIQ-7 scores (S1 Table).

## Criterion validity

The Spearman correlation coefficient between the subscale POPDI-6 score and the POP-SS-7 in the Amharic version was $r = 0.71$, $p<0.001$, demonstrating very good criterion validity for POPDI-6. Moreover, the SCC between the summary PFDI-20 and the POP-SS-7 score was $r = 0.67$, $p<0.001$, which proved that the POP-SS-7 score was significantly correlated with the POPDI-6 score (S2 Table).

## Construct validity: Known groups

There were statistically significant differences among the two groups of PFDs in the PFDI-20 and PFIQ-7 scores (Mann–Whitney U Test; *p < 0.001*). Women with POP had significantly higher total and subscale scores on PFDI-20 and PFIQ-7 compared to non-prolapse women ($p < 0.001$), especially in the POPDI-6 (116.4 ± 48.4 vs 2.2 ± 3.6; $p < 0.001$) and POPIQ-7 (60.6 ± 24.2 vs 1.1 ± 1 .9; $p < 0.001$). Likewise, total PFDI-20, PFIQ-7, and all subscales scores were higher in women with than without UI and AI ($p < 0.001$). The mean scores of the PFDI-20 and PFIQ-7 scales by PFDs are shown in Table 4.

The analysis of SCC confirmed that both summary and subscale of the PFDI-20 and PFIQ-7 scores were correlated with the POP-Q vaginal examination findings (SCC Test; *p < 0.001*). The POP-Q significantly correlated with PFDI-20 score (SCC = 0.69; $P < 0.001$) and with PFIQ-7 score (SCC = 0.71; $P < 0.001$). Furthermore, the PFIQ-7 score was significantly

**Table 4. Comparison of total and subscale PFDI-20 and PFIQ-7 scores between women with and without PFDs (known-groups validity).**

|  | POP | | | UI | | | AI | | |
|---|---|---|---|---|---|---|---|---|---|
|  | Yes | No | P value* | Yes | No | P value* | Yes | No | P value* |
| Number | 117 | 80 |  | 63 | 134 |  | 36 | 161 |  |
| PFDI-20 | 116.4±48.4 | 2.2±3.6 | <0.001 | 101.4±55.1 | 55.2±67.9 | <0.001 | 113.2±52.5 | 60.3±66.8 | <0.001 |
| POPDI-6 | 53.1±22.2 | 2.9±4.6 | <0.001 | 46.6±24.1 | 26.2±30.6 | <0.001 | 49.4±20.1 | 28.9±30.8 | <0.001 |
| CRADI-8 | 18.4±12.8 | 1.1±2.3 | <0.001 | 16.3±13.7 | 9.1±12.2 | <0.001 | 19.4±12.9 | 9.5±12.5 | <0.001 |
| UDI-6 | 40.2±25.0 | 2.2±4.5 | <0.001 | 33.9±24.9 | 20.5±26.9 | <0.001 | 43.8±24.1 | 20.5±28.1 | <0.001 |
| PFIQ-7 | 92.2±34.2 | 1.6±2.3 | <0.001 | 88.1±46.3 | 22.0±27.3 | <0.001 | 95.1±40.4 | 46.5±49.9 | <0.001 |
| UIQ-7 | 47.6±23.4 | 1.1±2.8 | <0.001 | 42.9±28.4 | 5.3±6.6 | <0.001 | 49.8±25.0 | 23.9±28.1 | <0.001 |
| CRAIQ-7 | 10.5±7.2 | 1.2±2.1 | <0.001 | 29.7±17.9 | 27.6±33.6 | <0.001 | 9.5±7.5 | 6.1±37.1 | <0.001 |
| POPIQ-7 | 60.6±24.2 | 1.1±1 .9 | <0.001 | 55.2±29.4 | 40.0±47.0 | <0.001 | 61.0±25.7 | 30.9±34.2 | <0.001 |

*PFDs* pelvic floor disorders, *POP* pelvic organ prolapse, *UI* urinary incontinence, *AI* anal incontinence, *PFDI-20* Pelvic Floor Distress Inventory–Short Form 20, *POPDI* Pelvic Organ Prolapse Distress Inventory, *CRADI* Colorectal–Anal Distress Inventory, *UDI* Urinary Distress Inventory, *PFIQ-7* Pelvic Floor Impact Questionnaire-7, *UIQ* Urinary Impact Questionnaire, *CRAIQ* Colorectal–Anal Impact Questionnaire, *POPIQ* Pelvic Organ Prolapse Impact Questionnaire, Mean ± standard deviation
*Mann–Whitney U test

correlated with PFDI-20 score (SCC = 0.79; P < 0.001). The higher SCC were related to POP dimensions, either in PFDI-20 (POPDI = 0.73) and PFIQ-7 (POPIQ = 0.72). See S3 Table.

All predefined hypotheses were confirmed, as shown in Table 4 and criterion validity. Women with POP reported higher scores on the POPDI-6 (116.4 ± 48.4) and POIQ-7 (60.6 ± 24.2) scales than the non-prolapse women (2.2 ± 3.6 and 1.1 ± 1 .9, respectively, p <0.001, Table 4). Similarly women with UI and FI reported higher scores on subscale PFDI-20 score (UDI-6: 33.9±24.9 vs 20.5±26.9 and CRADI-819.4±12.9 vs 9.5±12.5) and PFIQ-7 score (UIQ-7: 42.9±28.4 vs5.3±6.6 and CRAIQ-7: 9.5±7.5 vs 6.1±37.1) than their counterparts, respectively (Table 4). Moreover, PFDI-20 and PFIQ-7 scores are correlated with POP-Q scores (criterion validity).

## Discussion

PFDI-20 and PFIQ-7 have proven to be valid and reliable instruments to assess PFDs symptom distress and measure their impact on women's HRQoL [13, 14]. Until now, their Amharic translations have not been validated. In this study, we evaluated the reliability and validity of the Amharic versions of the PFDI-20 and PFIQ-7. Both instruments were successfully translated and culturally adapted. The results showed excellent internal consistency and test–retest reliability, acceptable construct and very good criterion validity. All assumptions were confirmed. Thus, the translated versions can be used as a standard tool in clinics and research in Amharic-speaking women with PFDs.

It is essential to validate the instrument in a new population and language before its use [25, 35]. And the validation requires several steps, including linguistic, cultural, and psychometric validation [25, 27]. In the present study, the linguistic translation and cultural adaptation were performed using a systematic approach. Since direct word-for-word translation does not guarantee sufficient equivalence, emphasis was given to maintaining the original context and meaning of the words of the questionnaires. Back-translation by two translators with different backgrounds was performed in consideration of the differences in medical terminology and subtle nuances. There were no changes in the instructions; and lay-out of the questionnaires. Moreover, the format of the instruments was the same as the original scale ensuring technical equivalence. Translations were similar to previous adaptations to Tigrigna, Swedish, Turkish

and Spanish [16, 17, 19]. Content validity was determined in a similar way as described by previous validation studies [16, 17, 19]. Accordingly, we found that the instruments are content valid. This was evidenced by excellent expert panel agreement on the relevance of items, being reviewed by multilingual expert translators, and acceptance of the instruments during pretesting.

The absence of difficulty in responding to the majority of items and the ease of completion within a short period partly provides evidence for the acceptability of the Amharic versions in the region in which the study was conducted. The time for completion in the present study was comparable to the Spanish, Swedish and French validations [17, 36, 37]. The average time of filling in was 9 min for the PFDI-20 Amharic version, very similar to the French (9.2 min) and Spanish (10.1 min) versions [36, 37]. Regarding the PFIQ-7 Amharic version, it was 7.0 min, higher than the French version (3.4 min), but similar to Spanish version (7.5 min) [36, 37].

This study establishes the reliability of Amharic versions of the PFDI-20 and PFIQ-7 questionnaires, as predicted. Both summary and subscale scores of the translated versions showed excellent internal consistency with Cronbach's alpha between 0.84 and 0.95. The same or very similar alphas were found in the Tigrigna [19], Polish [38], Greek [18], and Finnish [39] versions. The two-week test-retest reliability also demonstrated an excellent correlation between the paired test-retest scores of PFDI-20 (ICC of 0.97 in the PFDI-20 total score, and a range from 0.95 to 0.96 for its subscales) and PFIQ-7 (ICC of 0.86 for summary PFIQ-7 score and a range from 0.87 to 0.94 for subscales; $p < 0.001$). Our findings were higher compared with a previous study of Tigrigna [19], Polish [38], Greek [18], Finnish [39], and Norwegian [15] women. The good test–retest reliability guaranteed that the questionnaire results were consistent over time.

Although no gold standard for PFD symptoms, the correlation between PFDI-20 and POPDI-6, and POP-SS-7 were used to estimate criterion validity indirectly. The criterion validity was found to be within an acceptable range. These findings are consistent with those of previous studies [9, 13], although they used different relative standard as a criterion.

The current study also verified good construct validity, showing significant correlation between PFDI-20 and PFIQ-7 and objective vaginal examination findings. The result is similar to some validation studies [15, 40, 41]. The PFDI-20 and PFIQ-7 scores increased as the POP-Q increased. As most women with POP in our study were stage III or IV, PFDI-20 and PFIQ-7 total and subscale scores were higher than in other studies [40, 42, 43]. The possible reason for this might be population in our study was from rural areas and primarily uneducated, with significant barriers to accessing care, resulting in longstanding prolapse and influencing symptoms bother.

The strength of this study were the adoption of a multistep translation method, as supported by existing evidence, rather than the simple translation-back-translation process [25, 27]. However, some limitations must be considered when interpreting the results of our study. First, our study was conducted in an urban, single hospital; therefore, the results may not be generalizable to populations in rural and remote areas. Further validation studies in more general contexts are therefore recommended. Second, conducting a survey with self-report measures entails potential bias due to socially desirable responses. Specifically, rates of illiteracy may impact the validity (bias). Future studies should aim to utilize other methodologies that would enable a more in-depth analysis. Third, responsiveness to change was not evaluated. Since the ability to detect a change in prolapse symptoms due to an intervention is an important scale property, we recommend the inclusion of this in future studies. Fourth, it was not possible to compare construct validity with other generic questionnaires used for PFD evaluation, for example, the Short Form Health Survey (SF-12) and International Consultation on

Incontinence Questionnaire Short Form (ICIQ-SF). This is because of the absence of translated and validated versions in Amharic.

## Conclusion

The Amharic versions of PFDI-20 and PFIQ-7 achieved good semantic, conceptual, idiomatic and content equivalence with the original versions. The translated Amharic versions of the PFDI-20 and PFIQ-7 are reliable, valid and feasible to measure symptoms and their impact on HRQoL in Ethiopian Amharic-speaking women with PFDs. The questionnaires can be easily administered and used in research and clinical settings. The Amhara Regional Health Bureau should consider integrating these questionnaires into service delivery in the region. However, prior piloting and modification for wider applicability, especially outside the study area is needed. Further studies are also needed to evaluate the responsiveness of PFDI-20 and PFIQ-7.

## Supporting information

**S1 File. Amharic version of the short form of PFDI-20 and PFIQ-7.**
(PDF)

**S2 File. English version of the short form of PFDI-20 and PFIQ-7.**
(PDF)

**S1 Dataset. Minimal dataset.**
(SAV)

**S1 Table. Floor and ceiling effects of baseline PFDI-20 and PFIQ-7 scores.**
(DOCX)

**S2 Table. Correlation between POP-SS and PFDI-20 and PFIQ-7 scores (criterion validity).**
(DOCX)

**S3 Table. Spearman's correlation coefficient (SCC) between total and subscale scores of PFDI-20, PFIQ-7 and measures of pelvic examination (Pelvic Organ Prolapse Quantification).**
(DOCX)

## Author Contributions

**Conceptualization:** Tadesse Belayneh Melkie, Zelalem Mengistu Gashaw, Zelalem Ayichew Workineh, Tamiru Minwuye Andargie.

**Data curation:** Tadesse Belayneh Melkie, Zelalem Mengistu Gashaw, Zelalem Ayichew Workineh, Tamiru Minwuye Andargie, Tibeb Zena Debele.

**Formal analysis:** Tadesse Belayneh Melkie, Zelalem Mengistu Gashaw.

**Funding acquisition:** Zelalem Mengistu Gashaw, Zelalem Ayichew Workineh, Tamiru Minwuye Andargie.

**Investigation:** Tadesse Belayneh Melkie.

**Methodology:** Tadesse Belayneh Melkie, Zelalem Mengistu Gashaw, Zelalem Ayichew Workineh, Tamiru Minwuye Andargie, Solomon Gedlu Nigatu.

**Project administration:** Zelalem Mengistu Gashaw.

**Supervision:** Tamiru Minwuye Andargie, Tibeb Zena Debele.

**Validation:** Tadesse Belayneh Melkie, Zelalem Mengistu Gashaw, Zelalem Ayichew Workineh, Tibeb Zena Debele.

**Visualization:** Tadesse Belayneh Melkie, Zelalem Mengistu Gashaw, Zelalem Ayichew Workineh, Solomon Gedlu Nigatu.

**Writing – original draft:** Tadesse Belayneh Melkie.

**Writing – review & editing:** Tadesse Belayneh Melkie, Zelalem Mengistu Gashaw, Zelalem Ayichew Workineh, Tamiru Minwuye Andargie, Tibeb Zena Debele, Solomon Gedlu Nigatu.

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
