## [Decision Letter · Decision Letter 0]

31 Aug 2022

PONE-D-22-16526Translation, reliability, and validity of Amharic versions of the Pelvic Floor Distress Inventory (PFDI-20) and Pelvic Floor Impact Questionnaire (PFIQ-7)PLOS ONE

Dear Dr. Belayneh,

Thank you for submitting your manuscript to PLOS ONE. After careful consideration, we feel that it has merit but does not fully meet PLOS ONE’s publication criteria as it currently stands. Therefore, we invite you to submit a revised version of the manuscript that addresses the points raised during the review process.

Please find the reviewer comments below. Their suggestions are mostly related with minor issues. King Regards

We look forward to receiving your revised manuscript.

Kind regards,

Fatih Özden, PhD

Academic Editor

PLOS ONE

Journal Requirements:

Reviewers' comments:

Reviewer's Responses to Questions

**Comments to the Author**

1. Is the manuscript technically sound, and do the data support the conclusions?

Reviewer #1: Yes

Reviewer #2: Yes

2. Has the statistical analysis been performed appropriately and rigorously? 

Reviewer #1: Yes

Reviewer #2: Yes

3. Have the authors made all data underlying the findings in their manuscript fully available?

Reviewer #1: Yes

Reviewer #2: Yes

4. Is the manuscript presented in an intelligible fashion and written in standard English?

Reviewer #1: Yes

Reviewer #2: Yes

5. Review Comments to the Author

Reviewer #1: If you will also provide content validity, then very close result obtain. Because for this purpose of analysis two validity methods are essential construct and content. Also supplementary material required in English version.

Reviewer #2: The authors have translated, analyzed the reliability, and validity of Amharic versions of the Pelvic Floor Distress

Inventory (PFDI-20) and Pelvic Floor Impact Questionnaire (PFIQ-7). The paper is well written, with a good scientific soundness and logical flow. However, I have a few comments or observations below:

-Please check if numbers are correct in the following sentence "Among those 197 participants at baseline, 117 (59.4%) were diagnosed with POP and 96 (48.8%) 256 with POP-Q stage III and IV."

-How was the sample size calculated for the test-retest reliability? It needs to be further clarified.

6. PLOS authors have the option to publish the peer review history of their article (what does this mean?). If published, this will include your full peer review and any attached files.

Reviewer #1: **Yes: **Dr. Lalit K. Toke, Associate Professor, Mechanical Engineering Department, Sandip Institute of Engineering and Management, Nasik, India.

Reviewer #2: No

---

## [Decision Letter · Decision Letter 1]

28 Sep 2022

Translation, reliability, and validity of Amharic versions of the Pelvic Floor Distress Inventory (PFDI-20) and Pelvic Floor Impact Questionnaire (PFIQ-7)

PONE-D-22-16526R1

Dear Dr. Belayneh,

We’re pleased to inform you that your manuscript has been judged scientifically suitable for publication and will be formally accepted for publication once it meets all outstanding technical requirements.

Kind regards,

Fatih Özden, PhD

Academic Editor

PLOS ONE

Additional Editor Comments (optional):

Reviewers' comments:

Reviewer's Responses to Questions

**Comments to the Author**

1. If the authors have adequately addressed your comments raised in a previous round of review and you feel that this manuscript is now acceptable for publication, you may indicate that here to bypass the “Comments to the Author” section, enter your conflict of interest statement in the “Confidential to Editor” section, and submit your "Accept" recommendation.

Reviewer #1: All comments have been addressed

2. Is the manuscript technically sound, and do the data support the conclusions?

Reviewer #1: Yes

3. Has the statistical analysis been performed appropriately and rigorously? 

Reviewer #1: Yes

4. Have the authors made all data underlying the findings in their manuscript fully available?

Reviewer #1: Yes

5. Is the manuscript presented in an intelligible fashion and written in standard English?

Reviewer #1: Yes

6. Review Comments to the Author

Reviewer #1: Dear Author, Your work is good, also you have responded well to the reviewers comments. I hope your paper will get more readability.

7. PLOS authors have the option to publish the peer review history of their article (what does this mean?). If published, this will include your full peer review and any attached files.

Reviewer #1: **Yes: **Dr. Lalit K. Toke, Associate Professor, Mechanical Engineering Department, Sandip Institute of Engineering and Management, Nashik, Maharashtra, India

---

## [Editor Report · Acceptance letter]

9 Nov 2022

PONE-D-22-16526R1 

Translation, reliability, and validity of Amharic versions of the Pelvic Floor Distress Inventory (PFDI-20) and Pelvic Floor Impact Questionnaire (PFIQ-7) 

Dear Dr. Melkie:

I'm pleased to inform you that your manuscript has been deemed suitable for publication in PLOS ONE. Congratulations! Your manuscript is now with our production department. 

Kind regards, 

on behalf of

Dr. Fatih Özden 

Academic Editor

PLOS ONE